# Evaluation of the Reliability and the Performance of Magnetic Resonance Imaging Radiomics in the Presence of Randomly Generated Irrelevant Features for Prostate Cancer

**DOI:** 10.3390/diagnostics13233580

**Published:** 2023-12-01

**Authors:** Cindy Xue, Jing Yuan, Gladys G. Lo, Darren M. C. Poon, Winnie C. W. Chu

**Affiliations:** 1Research Department, Hong Kong Sanatorium and Hospital, Hong Kong, China; cindyxue@link.cuhk.edu.hk (C.X.);; 2Department of Imaging and Interventional Radiology, The Chinese University of Hong Kong, Hong Kong, China; 3Department of Diagnostic and Interventional Radiology, Hong Kong Sanatorium and Hospital, Hong Kong, China; 4Comprehensive Oncology Centre, Hong Kong Sanatorium and Hospital, Hong Kong, China

**Keywords:** radiomics, reliability, random features, prostate cancer, MRI, machine learning

## Abstract

Radiomics has the potential to aid prostate cancer (PC) diagnoses and prediction by analyzing and modeling quantitative features extracted from clinical imaging. However, its reliability has been a concern, possibly due to its high-dimensional nature. This study aims to quantitatively investigate the impact of randomly generated irrelevant features on MRI radiomics feature selection, modeling, and performance by progressively adding randomly generated features. Two multiparametric-MRI radiomics PC datasets were used (dataset 1 (*n* = 260), dataset 2 (*n* = 100)). The endpoint was to differentiate pathology-confirmed clinically significant (Gleason score (GS) ≥ 7) from insignificant (GS < 7) PC. Random features were generated at 12 levels with a 10% increment from 0% to 100% and an additional 5%. Three feature selection algorithms and two classifiers were used to build the models. The area under the curve and accuracy were used to evaluate the model’s performance. Feature importance was calculated to assess features’ contributions to the models. The metrics of each model were compared using an ANOVA test with a Bonferroni correction. A slight tendency to select more random features with the increasing number of random features introduced to the datasets was observed. However, the performance of the radiomics-built models was not significantly affected, which was partially due to the higher contribution of radiomics features toward the models compared to the random features. These reliability effects also vary among datasets. In conclusion, while the inclusion of additional random features may still slightly impact the performance of the feature selection, it may not have a substantial impact on the MRI radiomics model performance.

## 1. Introduction

Radiomics is a field of medical imaging that encompasses the extraction and analysis of quantitative features from medical images. Radiomics offers the potential to unveil imaging patterns and biomarkers that may help improve disease characterization, treatment selection, and the monitoring of treatment response [1,2]. Prostate cancer (PC) is one of the most prevalent cancers in men, particularly affecting older men [3]. The incidence rates of PC vary across countries and populations, with higher rates observed in African-American men [4]. Multiple risk factors, including age, family history, genetic mutation, and obesity, contribute to the development of PC [3,4,5]. Recent advancements in early detection and treatment options, such as radiotherapies, targeted therapies, and immunotherapies, have contributed to improved mortality rates in PC patients [4]. Given the heterogeneous nature of PC and the varying responses to treatment observed among patients, an accurate assessment of its aggressiveness is of utmost importance in determining appropriate treatment strategies [3,5]. By understanding the specific characteristics of PC, including its growth rate and potential for metastasis, healthcare professionals can tailor treatment plans to maximize efficacy while minimizing unnecessary interventions. The Gleason score, a histopathological grading system, has traditionally been used for assessing PC aggressiveness [6]. It assesses the architectural patterns of the tumor cells obtained through biopsy or surgery and assigns a grade ranging from 1–5 to the two most prevalent patterns observed. The sum of these two grades represents the Gleason score, with higher scores indicating more aggressive cancer [6]. While the Gleason score has been an important metric in PC diagnosis and treatment planning, it has some limitations. The Gleason score relies on subjective interpretation by pathologists and can be affected by sampling errors due to the heterogeneity of prostate tumors. This has led to an increased interest in incorporating other measurements, such as the utilization of imaging-based information, to enhance the accuracy and reliability of PC assessment. The Prostate Imaging–Reporting and Data System (PI-RADS) and Likert scoring system were proposed and developed to standardize the interpretation of prostate MRI and quantify radiologists’ confidence in the presence of PC, respectively [7]. Both scoring systems were shown to have high detection rates of clinically significant PC, which could complement the Gleason score in PC assessment, but they relied too much on readers’ experience and were subject to inter-observer disagreement [7]. Quantitative MRI analysis, such as MRI radiomics, has been actively investigated and reported to aid in the objective detection and grading of PC [8,9,10].

Although MRI radiomics studies have shown great promise, there is a concern regarding the reliability of radiomics [11,12,13]. This issue requires further investigation to address various unresolved issues and ensure the accuracy and dependability of radiomics-based approaches in practical healthcare applications. These reliability issues could stem from many contributing factors, which could happen during image acquisition, reconstruction, feature extractions, and other procedures [14,15,16]. Furthermore, radiomics studies normally involve an extremely large number of features, which could pose challenges in data analysis and interpretation. Having more features/data could intuitively provide more information, which could aid in building more comprehensive and better models, which consequently help improve the predictive performance of the models [17]. However, having a large number of features could also result in a high-dimensional dataset or the inclusion of unwanted or irrelevant features. While some radiomics features generated may contribute to predicting or understanding the clinical endpoint references, not all features are deemed important or highly associated with the clinical endpoint references, which could become irrelevant features. These irrelevant features could potentially overshadow or dilute the real valuable radiomics information, which can lead to inaccurate and inefficient results. Using a high-dimensional dataset is also prone to overfitting, increased computational complexity, and further decreased interpretability of the results [18].

Some machine learning techniques were used to mitigate this situation, which include, but are not limited to, feature selections and data modeling. Feature selection methods could discern the most critical features and eliminate the redundant and irrelevant features from the large amount of data extracted from the medical images. These features would then be fitted into models via data modeling techniques, which could predict or classify the patients [19,20]. Recursive feature elimination (RFE), minimum redundancy maximum relevance (MRMR) feature selection, and least absolute shrinkage and selection operator (LASSO) are some of the most commonly used feature selection methods, while random forest (RF) and LASSO modeling are commonly applied technique in MRI radiomics modeling [19,20,21,22,23]. However, it is unclear how this high-dimensional dataset with the inclusion of irrelevant features will affect this machine learning technique (especially feature selection or modeling) in building a robust MRI radiomics model. Hence, this study aims to quantitatively investigate and evaluate the impact of random features, which are completely irrelevant to the endpoints on MRI radiomics feature selection, modeling, and performance.

## 2. Materials and Methods

This study employed publicly accessible radiomics feature datasets, which facilitated reproducibility and external validation. Consequently, the retrospective nature of the study allowed for the waiver of Institutional Review Board (IRB) approval.

### 2.1. Datasets

Two publicly available MRI prostate radiomics feature datasets were used in this study (Table 1). The clinical outcome of both datasets was divided into clinically significant PC (Gleason score ≥ 3 + 3) and non-clinically significant PC (Gleason score = 3 + 3). These outcomes are lesion-based, with one derived from MR-guided biopsy [21] and another from whole-mount prostatectomy [22]. Both methods were mentioned in the previous literature [21,22]. Only the radiomics feature values provided in the original studies were used in this study. Z-score normalization was conducted to normalize the radiomics features. The feature names used in this study are shown in Appendix A.

#### Training Dataset and Testing Dataset

The data from each dataset were separated into training and hold-out testing datasets with a ratio of 7:3.

### 2.2. The Addition of Randomly Generated Irrelevant Features

The randomly generated irrelevant features (random features) were added by randomly fabricated features using the uniform distribution with the range of randomly picked radiomics features of the dataset for each new feature. The details of the random feature generation are described in Appendix A. The number of randomly generated features was added at 12 levels with an increment of 10% and an addition of a 5% increment (0%, 5%, 10%, 20%, 30%, 40%, 50%, 60%, 70%, 80%, 90%, and 100%) and were repeated 20 times at each level to increase the accuracy and the reliability of the results. Each repetition was conducted with a different random sample selection of the radiomics features, where its range will be mimicked to generate the random features. 

### 2.3. Radiomics Feature Selection Methods

The Mann-Whitney U test (with a 0.05 threshold) was conducted to filter the radiomics features. After filtering, 3 different feature selection methods, which were commonly used in radiomics studies [21,22,23], were used to select 20 and 50 features at maximum. They were recursive feature elimination (RFE), LASSO and minimum redundancy maximum relevance (MRMR). The parameter settings of the feature selections are listed in Appendix A.

### 2.4. Machine Learning Classifiers

After passing through feature selection, two classifiers were utilized to build the models: random forest (RF) classifiers and LASSO. Finally, 12 radiomics models (3 feature selection algorithms × 2 classifiers × 2 maximum threshold) were built for each reference permutation. They were LASSO + RF, RFE + RF, MRMR + RF, RFE + LASSO, MRMR + LASSO, and LASSO + LASSO for each maximum threshold (20 and 50). The parameter settings of the classifiers are listed in Appendix A.

### 2.5. Data Training, Validation, and Testing

Dataset 1 was trained using ten-fold stratified cross-validation, while dataset 2 was trained using five-fold stratified cross-validation due to its relatively smaller sample size. Oversampling was conducted to help address the data imbalance. The performances of the radiomics models trained with or without the additional random features were first assessed and compared. True performances of these radiomics models were also evaluated using the hold-out testing datasets.

### 2.6. Features Importance

Feature importance was calculated to evaluate the contribution of each feature to the model’s performance. To calculate the distribution of contribution of the features in the model built using RF classifiers, the varImp function in R was used, which determines feature importance by assessing the reduction in model performance when each predictive feature is randomly permuted while keeping other features unchanged. For models built using LASSO, the feature importance is denoted using the absolute value of the magnitude of the coefficients obtained from the LASSO models. The obtained feature importance values were further scaled to a 0–100 percentage range for ease of interpretation.

The importance of the top 10 radiomics features was calculated by averaging the importance of each feature across all repetitions and different feature selection methods for different ratios of additional random features. Similarly, the importance of the top 10 random features was calculated by averaging the 10 highest importance of random features combined across all repetitions and different feature selection methods for the different ratios of additional random features.

### 2.7. Statistics

The Jaccard Similarity Coefficient (JSC) was used to evaluate the consistency of the radiomics feature selected by different feature selection methods (Formula (1)) [24]. The JSC ranges from 0 to 1, with 0 representing no feature commonly selected and 1 representing the exact same features selected. Since there were 20 repetitions in each permutation level, the JSC was calculated by the intersection of radiomics features selected from all 20 different permuted datasets over the union of those datasets (Formula (2)).
(1)$$JSC=\frac{A\cap B}{A\cup B}$$
(2)$$JSC=\frac{D_{1}\cap D_{2}\cap \ldots D_{20}}{D_{1}\cup D_{2}\cup {\ldots D}_{20}}$$

A = Dataset AB = Dataset BD_i_ = Dataset i (with i representing the number of repetitions)

The area under the curve (AUC) of the Receiver Operating Characteristics (ROCs) curve, sensitivity, specificity, and accuracy of each model were computed and compared using an ANOVA test with a Bonferroni correction [25]. A significance level of a *p*-value < 0.05 was utilized to determine statistical significance. These statistical analyses were performed in R Studio.

## 3. Results

### 3.1. Feature Selection

The number of random features selected by each feature selection varied across different levels, with additional random features added to the dataset. The results of the three different feature selection methods with different maximum thresholds (20 and 50 features selected) in selecting random features are shown in Figure 1. Overall, there is an increasing trend of random features selected by feature selection methods with an increasing number of additional random features. LASSO feature selection selected the most random features from both datasets, followed by MRMR, which selected at most 55% (*n* = 11) when the maximum threshold = 20 and 36% (*n* = 18) when the maximum threshold is 50 in dataset 2. Meanwhile, compared to other feature selections, RFE relatively selected the fewest random features for both maximum thresholds in both datasets, especially when the ratio of random features added into the datasets was equal to or greater than 50% (≥50%) (*p* < 0.05). The ratio is also higher when the threshold is 20 compared to when the threshold is 50, which means the number of random features selected is comparable for both thresholds.

The most stable features selected using different feature selection methods in each dataset are shown in Appendix A. The JSC calculated to compare the selected radiomics features in all 20 repetitions by every feature selection method across different levels of additional random features to the dataset was represented in Figure 2. MRMR selected the most consistent features during the 20 repetitions in both datasets, with better performance in dataset 1 than dataset 2, while LASSO and RFE performed comparably. There is a slight decrease in the JSC with the increase in the number of random features added to the datasets.

### 3.2. Model Performances

#### 3.2.1. Area under the Curve (AUC)

The AUCs in the training sets were not significantly different among different combinations of the feature selection and data modeling technique across different numbers of random features added to the dataset (mean AUC_training set_ without the additional random features = 0.96; 0.97 for dataset 1 and dataset 2 consecutively; mean AUC_training set_ with random features = 0.95; 0.95 for dataset 1 and 2 consecutively; *p* > 0.05). AUCs in the testing set were also not significantly different among the different combinations of feature selection and data modeling technique (mean AUC_testing set_ without the additional random features = 0.91; 0.72 for dataset 1 and dataset 2; mean AUC_testing set_ with additional random features = 0.93; 0.67 for dataset 1 and dataset 2; *p* > 0.05) despite different numbers of random features added into the dataset, as shown in Figure 3.

#### 3.2.2. Model Accuracy

The accuracy trend in both training and testing sets was similar to the trend of the performance measured by the AUC, with no significant difference among the different combinations of feature selection and data modeling technique despite different numbers of random features added to the dataset (*p* > 0.05), as shown in Figure 4.

The accuracy measurements of the models in the training sets were not significantly different across different numbers of random features added to the dataset (mean accuracy_training set_ without the additional random features = 0.94; 0.96 for dataset 1 and dataset 2 consecutively; mean accuracy_training set_ with random features = 0.91; 0.90 for dataset 1 and 2 consecutively; *p* > 0.05). Accuracy measurements in the testing set were also not significantly different despite different numbers of random features added to the dataset (mean accuracy_testing set_ without the additional random features = 0.91; 0.81 for dataset 1 and dataset 2; mean accuracy_testing set_ with additional random features = 0.93; 0.79 for dataset 1 and dataset 2; *p* > 0.05). The sensitivity and specificity of the models are shown in Appendix A.

#### 3.2.3. Feature Importance

Feature importance was calculated for all the models. Figure 5 shows the importance of the top 10 most important features (both radiomics and random) along with the total mean contribution of random features for dataset 1 and dataset 2 when the maximum threshold was set to 20. The figures of the top 10 most important features (radiomics and random features) toward the model, with the maximum threshold set to 50 for the feature selection methods, are shown in Appendix A. Overall, the importance of the random features is significantly lower for the top-performing random radiomics features (maximum (average): 0.47–1.20%) compared to the importance of the top radiomics features (maximum (average): 8.32–22.64%) (*p* < 0.05).

## 4. Discussion

With its inherently large size and complexity, the radiomics dataset inevitably may include random features that are irrelevant to the clinical endpoint references and could pose significant challenges for data analysis. This could affect the accuracy, efficiency, and reliability of the performance of the built model. This study investigated the impact of these random irrelevant features on MRI radiomics, modeling, and performance by progressively adding randomly generated features to MRI radiomics datasets. Radiomics features were assumed to have more association with the clinical endpoint references, while the random features were deemed purely irrelevant to the clinical endpoint references.

This study showed that there is a slight tendency to select more random features with the increasing number of random features introduced to the datasets, especially when using LASSO feature selection. The ratio of the selected random features is higher when the maximum threshold is set to 20 than when it was set to 50. This decreasing ratio, despite the increase in the maximum threshold, indicated that there was no significant increase in the number of random features selected, even with a higher threshold for feature selection. The reliability of the feature selection measured by the JSC with 20 repetitions also did not change with the increase in the maximum threshold. This finding showed that the threshold set for the feature selection might not substantially affect the feature selection’s reliability and the number of random features selected. Overall, the results showed that additional random features, although irrelevant, could impact the reliability of the feature selection, with its influence increasing with the number of random features introduced. Some feature selection methods, such as MRMR, were also shown to be more reliable than others. However, this reliability effect also varies among datasets, where it might perform better in certain datasets than others.

Although the feature selection methods selected some random features for data modeling, the performance of the radiomics-built model did not seem to be especially impacted. Generally, the models performed relatively well and similarly across different numbers of random features introduced to the datasets. There are some factors that could result in this. Firstly, the number of random features selected after the feature selection was relatively few, especially when the additional random feature level was low. Secondly, the potentially strong performance of the modeling classifiers contributes to the building of an effective classification model. In addition, from the feature importance calculation, which assesses the contribution of the features to the radiomics models, the results showed that radiomics features generally were regarded as more important for the models compared to random features. The feature importance calculation showed that the radiomics model could identify the radiomics features, which could have a stronger association with the clinical endpoint references. Hence, when random features were introduced, their impact on the model’s performance was limited. These findings further highlighted the important and unique value of radiomics features in predicting the outcome and, parallelly, the robustness of these predictive model classifiers, as well as their abilities to discern meaningful patterns amidst random features.

In dataset 1, the performance of the models with the testing datasets was even slightly better when there were random features in the datasets than when there were none. These additional random features might also increase the diversity of the training data, which could improve the performance of the models [16,26]. On the other hand, in dataset 2, the model’s performance deteriorated slightly after the introduction of the additional random features. This finding showed that adding random features to this dataset might have introduced additional irrelevant information, causing the model to be less accurate. This discrepancy suggests that the impact of additional random feature information on model performance could still depend on the specific characteristics and complexity of the dataset. Hence, it emphasizes the importance of dataset-specific analysis and cautions against generalizing the effects of random features on model performance. However, the overall effect of the additional random features added to the dataset is minimal.

In the era of data-driven analyses and the increasing popularity of radiomics modeling, recent clinical models have increasingly utilized complex and high-dimensional data. This approach aims to capture a more comprehensive representation of the state of the disease, thereby enhancing the predictive capabilities of the models [17,27,28]. Essentially, this suggests that more data may lead to improved performance of the model. A study conducted by Gentile F et al. showed that by incorporating the prostate health index into multiparametric MRI, the resulting model not only exhibited superior classification performance of clinically significant PCs to non-clinically significant ones but also demonstrated greater reliability and repeatability compared to models using either multiparametric MRI or the prostate health index alone [28].

However, it is also important to acknowledge that the increasing number and complexity of data can cause the models to be susceptible to irrelevant information, which may consequently impact their performance [18]. This study showcases the potential of MRI radiomics-built models, which encompass complex and high-dimensional original radiomics datasets, including random features that are completely irrelevant to the clinical endpoint of differentiating clinically significant PCs from clinically insignificant ones. Remarkably, the overall performance of these MRI radiomics models, though data-dependent, is relatively robust in the presence of random features. This finding highlights the great potential of both radiomics features and machine learning modeling techniques in building clinical models, particularly for prostate cancer classification. The findings from this study could enhance confidence in the applicability of MRI radiomics-built models for clinical purposes.

Despite its findings, this study has several limitations. Firstly, the study design was retrospective and simulation-based, which could introduce selection bias, as the dataset may not represent the entire population or have incomplete information. The simulation nature of the study that tries to mimic real-world scenarios also might not fully capture the complexities and variability of actual clinical settings. Furthermore, it should be noted that this study only focused on generating uniformly distributed random features. It is important to acknowledge that different ways of generating random features could possess varying degrees of influence on the obtained results, which could be further evaluated in future studies. Secondly, using the public radiomics dataset in this study offered advantages in terms of reproducibility and external validation. However, it lacks image data, which prevents the visual demonstration of specific imaging features or patterns associated with the analyses. Consequently, the interpretation and discussion of results are solely based on quantitative radiomics data. In addition, this study only selected MRI PC datasets. Hence, the results might not be directly transferable to other organs, diseases, or imaging modalities. Moreover, PC often coexists with other prostatic conditions, such as prostatic hyperplasia or prostatitis [4]. The presence of these coexisting diseases could impact the performance of the MRI radiomics models in classifying PC, which was not part of this study. Further research is warranted to extend the study to PC with coexisting prostatic disease, other diseases, or imaging modalities. Lastly, the use of only three feature selection methods and two classifiers might not represent the broad range of available machine learning algorithms used in radiomics. However, we believe that the use of these commonly used methods in radiomics is deemed to be sufficient to give insights into the study focus. The optimization and comparative performance using different machine learning techniques are out of the scope of this study.

## 5. Conclusions

In conclusion, while the inclusion of additional random features, although irrelevant, could affect the feature selection process, it may not substantially impact the MRI radiomics model performance. It is still crucial to carefully consider the characteristics of the dataset and the relevance of the additional features added to the original datasets to build a robust and reliable MRI radiomics model. Future research can further explore advanced feature selection and data modeling techniques to ensure the inclusion of important/contributing features while mitigating the adverse effects of additional random features.

## Figures and Tables

**Figure 1 diagnostics-13-03580-f001:**
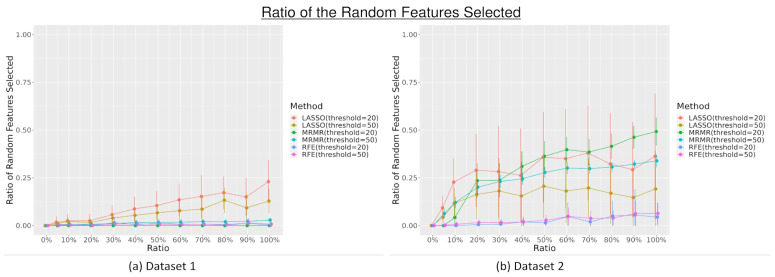
The ratio of the random features selected by the feature selection (least absolute shrinkage and selection operator (LASSO), minimum redundancy maximum relevance (MRMR), and recursive feature elimination (RFE)) with different maximum thresholds across different levels of additional random features.

**Figure 2 diagnostics-13-03580-f002:**
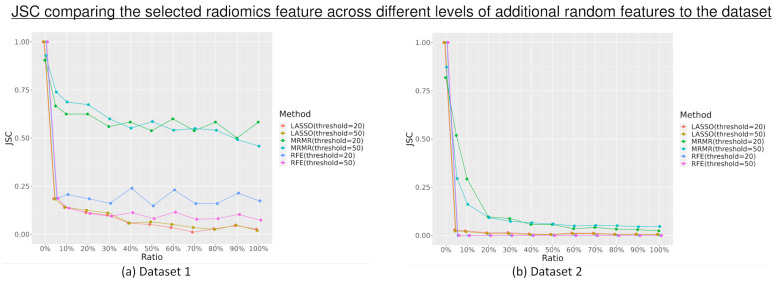
The Jaccard Similarity Coefficient (JSC) calculation comparing the selected radiomics features in all 20 repetitions by every feature selection method (least absolute shrinkage and selection operator (LASSO), minimum redundancy maximum relevance (MRMR), and recursive feature elimination (RFE)) across different levels of additional random features for the dataset.

**Figure 3 diagnostics-13-03580-f003:**
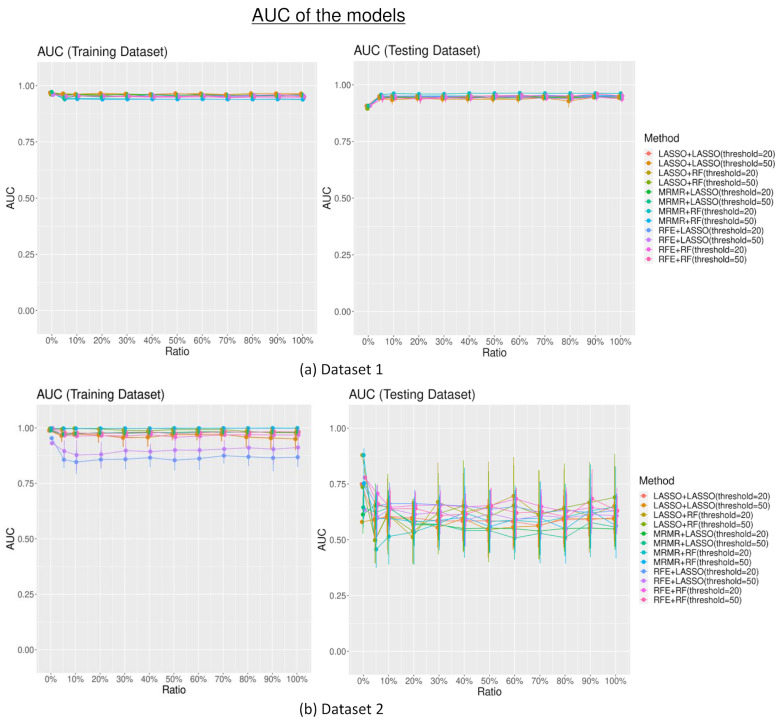
The area under the curve (AUC) performance of the models with different combinations of feature selection (least absolute shrinkage and selection operator (LASSO), minimum redundancy maximum relevance (MRMR), recursive feature elimination (RFE)), and classifiers (LASSO and random forest (RF)).

**Figure 4 diagnostics-13-03580-f004:**
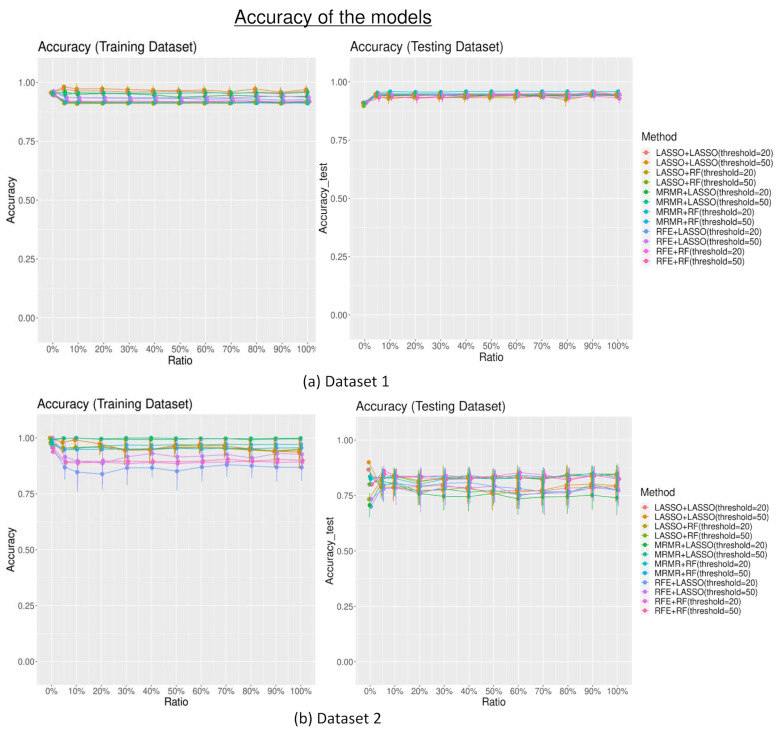
The accuracy of the models with different combinations of feature selection (least absolute shrinkage and selection operator (LASSO), minimum redundancy maximum relevance (MRMR), recursive feature elimination (RFE)), and classifiers (LASSO and random forest (RF)).

**Figure 5 diagnostics-13-03580-f005:**
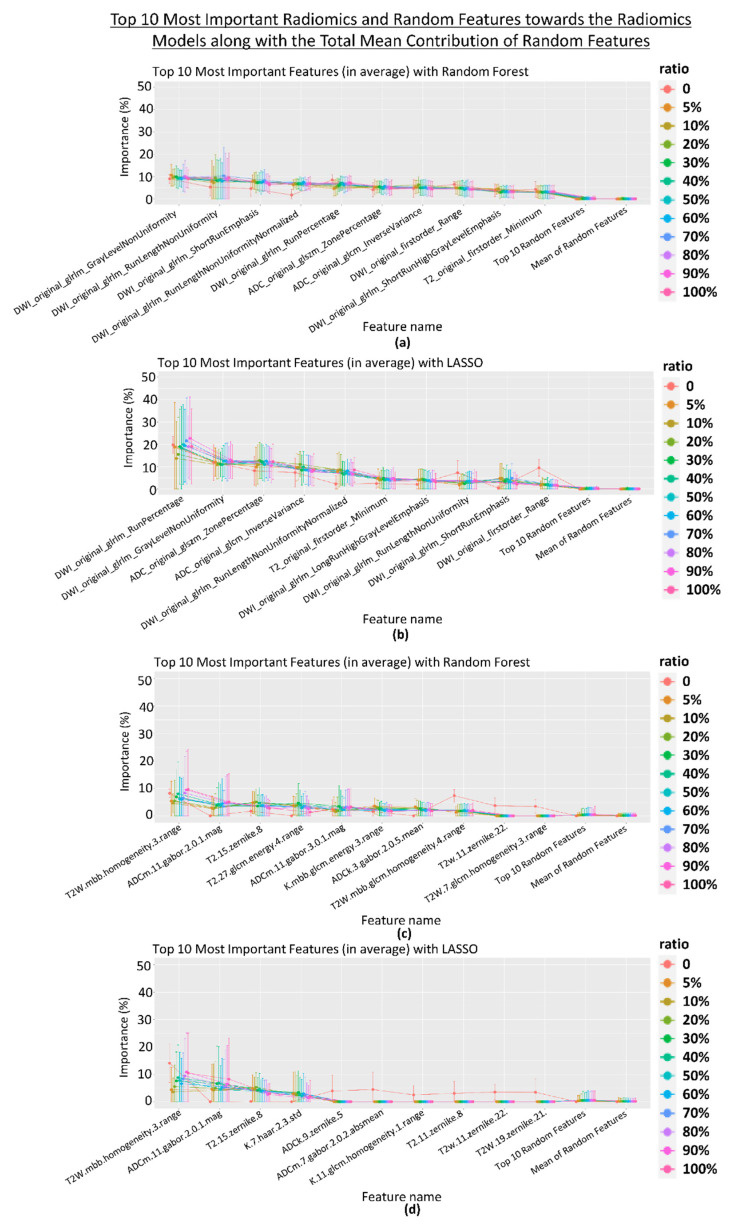
The importance of the top 10 most important radiomics and random features toward the radiomics model along with the total mean contribution of random features using random forest (RF) and least absolute shrinkage and selection operator (LASSO) classifiers for dataset 1 (**a**,**b**), respectively, and dataset 2 (**c**,**d**), respectively, with the maximum threshold set to 20 for the feature selection methods across every repetition and different feature selection methods for the different ratios of additional random features.

**Table 1 diagnostics-13-03580-t001:** Overview of the PC datasets used in this study.

Dataset	Sample Size	No. of Features	Outcome	Outcome Balance (%)	Modality	DOI
Song (dataset 1) [21]	260 lesions	265	clinically significant vs. non-clinically significant PC	49	Multiparametric MRI (T2W, DWI, and ADC maps)	https://doi.org/10.1371/journal.pone.0237587 (accessed on 12 April 2023)
Toivonen (dataset 2) [22]	100 lesions	7106	clinically significant vs. non-clinically significant PC	80	T2-TSE, DWI, and T2 mapping	https://doi.org/10.1371/journal.pone.0217702 (accessed on 12 April 2023)

## Data Availability

The study subjects or cohorts have been previously reported in the studies below (as this study uses publicly available data) [21,22].

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
