# Peer review of "Evaluation of the Reliability and the Performance of Magnetic Resonance Imaging Radiomics in the Presence of Randomly Generated Irrelevant Features for Prostate Cancer"

_diagnostics, 2023, doi:10.3390/diagnostics13233580_

Round 1

Reviewer 1 Report

Comments and Suggestions for Authors

1- The resolution of all of the figures is not clear and should be improved.

2- All equations should be numbered and referred to the corresponding number in the text.

3- Provide demographic information of patients in the text.

4- The manuscript has no appropriate description of the results. All figures should be described and data should be compared statistically.

5- Figure legends should be provided self-explanatory in detail. Also, all abbreviations used in tables should be defined in figure legends.

6- Moderate editing of English language grammar and spelling is required.

Comments on the Quality of English Language

Moderate editing of English language grammar and spelling is required.

Author Response

Response letter to reviewers

The authors would like to thank the reviewers for their thoughtful and in-depth review of our manuscript. In addressing these insightful comments, we believe that the manuscript has been strengthened considerably. In order to assist the review of our manuscript, the individual comments of each reviewer have been segmented, and followed by our response. Any change made to the original manuscript has been highlighted in the revised version of the manuscript.

Reviewer’s Comments

Comments and Suggestions for Authors

1- The resolution of all of the figures is not clear and should be improved.

  1. i) Text of the original criticism:

Throughout the manuscript

  1. ii) Author’s Response:

Thank the reviewer for his/her suggestion on the figures' resolutions. All of the figures have been improved.

2- All equations should be numbered and referred to the corresponding number in the text.

  1. i) Text of the original criticism:

Line 163-173

  1. ii) Author’s Response:

Thank the reviewer for his/her suggestion on the equation. All of the equations were numbered and referred to the corresponding number in the text.

3- Provide demographic information of patients in the text.

  1. i) Text of the original criticism:

Line 96-110

  1. ii) Author’s Response:

Thank the reviewer for his/her suggestion. Due to the nature of this study which uses public datasets, some of the demographic information of the patients was not available. We have added some information in Table 1 describing the dataset (line 110) and provided the reference information if more information is needed.

4- The manuscript has no appropriate description of the results. All figures should be described, and data should be compared statistically.

  1. i) Text of the original criticism:

Results section

  1. ii) Author’s Response:

Thank the reviewer for his/her suggestion. The results section has been improved accordingly.

5- Figure legends should be provided self-explanatory in detail. Also, all abbreviations used in tables should be defined in figure legends.

  1. i) Text of the original criticism:

Throughout the manuscript

  1. ii) Author’s Response:

Thank the reviewer for his/her suggestion. All of the figures' captions have been improved, and the abbreviations used were also listed.

6- Moderate editing of English language grammar and spelling is required.

  1. i) Text of the original criticism:

Throughout the manuscript

  1. ii) Author’s Response:

Thank the reviewer for his/her suggestion on the equation. Careful grammatical review and proofreading have been conducted to improve the language in the manuscript.

Reviewer 2 Report

Comments and Suggestions for Authors

Radiomics represents one of the most promising fields of cancer research. It is now possible, through artificial intelligence (AI) and machine learning (ML) advanced algorithms, to extract abundant quantitative features from patients scans and to analyze the high amount of data coming from these novel diagnostic tools to ultimately improve the non-invasive risk stratification and disease management of patients with prostate cancer (PCa).

This study aims to quantitatively investigate the impact of randomly generated irrelevant features on MRI radiomics feature selection, modeling, and performance by progressively adding randomly-generated features

I believe that the study has sufficient merit to be considered for publication, although major revisions are required. However, the authors should check the language in some points, and graphics should be more easily readable. It is lacking in some points that would add value to the entire manuscript:

Introduction.

The authors should add more information about PCa epidemiology as well done in this paper: https://doi.org/10.1159%2F000509434

Discussion.

Despite the many limitations allowed in the discussion, authors should clarify the real and/or potential benefits this method can add to clinical practice. Since no prospective study has been published so far, there is no reason at this time for these methods to be used in clinical practice. Authors should discuss how this model can fill the gap that still exists in this regard. Some ideas can be offered by reading this article: https://doi.org/10.1016/j.clgc.2022.04.013.

The authors should also discuss how the presence of coexisting prostatic disease should affect the accuracy of these methods. 

Comments on the Quality of English Language

Minor editing

Author Response

Response letter to reviewers

The authors would like to thank the reviewers for their thoughtful and in-depth review of our manuscript. In addressing these insightful comments, we believe that the manuscript has been strengthened considerably. In order to assist the review of our manuscript, the individual comments of each reviewer have been segmented, and followed by our response. Any change made to the original manuscript has been highlighted in the revised version of the manuscript.

Reviewer’s Comments

Radiomics represents one of the most promising fields of cancer research. It is now possible, through artificial intelligence (AI) and machine learning (ML) advanced algorithms, to extract abundant quantitative features from patients scans and to analyze the high amount of data coming from these novel diagnostic tools to ultimately improve the non-invasive risk stratification and disease management of patients with prostate cancer (PCa).

This study aims to quantitatively investigate the impact of randomly generated irrelevant features on MRI radiomics feature selection, modeling, and performance by progressively adding randomly-generated features

I believe that the study has sufficient merit to be considered for publication, although major revisions are required. However, the authors should check the language in some points, and graphics should be more easily readable. It is lacking in some points that would add value to the entire manuscript:

  1. i) Text of the original criticism:

Throughout the manuscript

  1. ii) Author’s Response:

Thank the reviewer for his/her suggestion on the manuscript editing and the graphics. Careful grammatical review and proofreading have been conducted to improve the language in the manuscript. In addition, all of the figures have been improved.

Introduction.

The authors should add more information about PCa epidemiology as well done in this paper: https://doi.org/10.1159%2F000509434. 

  1. i) Text of the original criticism:

Introduction section

  1. ii) Author’s Response:

Thank the reviewer for his/her suggestion on the introduction. We have expanded the introduction section to further describe the Pca epidemiology (lines 37 – 44)

Discussion.

Despite the many limitations allowed in the discussion, authors should clarify the real and/or potential benefits this method can add to clinical practice. Since no prospective study has been published so far, there is no reason at this time for these methods to be used in clinical practice. Authors should discuss how this model can fill the gap that still exists in this regard. Some ideas can be offered by reading this article: https://doi.org/10.1016/j.clgc.2022.04.013.

The authors should also discuss how the presence of coexisting prostatic disease should affect the accuracy of these methods. 

  1. i) Text of the original criticism:

Discussion section

  1. ii) Author’s Response:

Thank the reviewer for his/her suggestion on the discussion. We have added the discussion for the real and/or potential benefits this method can add to clinical practice (lines 330- 345). We have also added the discussion about the presence of coexisting prostatic disease in affecting the accuracy of these models (359-363).

Reviewer 3 Report

Comments and Suggestions for Authors

Summary

In this paper authors quantitatively investigated the impact of randomly generated irrelevant features on MRI radiomics feature selection, modelling, and performance by progressively adding randomly generated features. They used 2 public radiomics datasets of prostate cancer (PC), Dataset1 (n=260), and Dataset2 16 (n=100) to build the model and progress the features analysis and found that additional random features may not have a substantial impact on the MRI radiomics model but could affect the feature selection process. The paper was nicely organized and written well.

Comments

1.     There is no Table 1 which mentioned in Line 90.

2.     Figure 5 the feature name is not readable.

3.     Supplementary figure1 is the frame overlapped and can’t read.

Author Response

Response letter to reviewers

The authors would like to thank the reviewers for their thoughtful and in-depth review of our manuscript. In addressing these insightful comments, we believe that the manuscript has been strengthened considerably. In order to assist the review of our manuscript, the individual comments of each reviewer have been segmented, and followed by our response. Any change made to the original manuscript has been highlighted in the revised version of the manuscript.

Reviewer’s Comments

Summary

In this paper authors quantitatively investigated the impact of randomly generated irrelevant features on MRI radiomics feature selection, modelling, and performance by progressively adding randomly generated features. They used 2 public radiomics datasets of prostate cancer (PC), Dataset1 (n=260), and Dataset2 16 (n=100) to build the model and progress the features analysis and found that additional random features may not have a substantial impact on the MRI radiomics model but could affect the feature selection process. The paper was nicely organized and written well.

Comments

1.There is no Table 1 which mentioned in Line 90.

  1. i) Text of the original criticism:

Line 110

  1. ii) Author’s Response:

Thank the reviewer for his/her comment. Table 1 is now added to the manuscript.

  1. Figure 5 the feature name is not readable.
  2. i) Text of the original criticism:

Figure 5

  1. ii) Author’s Response:

Thank the reviewer for his/her suggestion on Figure 5. A new figure with a larger font is now uploaded.

  1. Supplementary figure1 is the frame overlapped and can’t read.
  2. i) Text of the original criticism:

Supplementary document

  1. ii) Author’s Response:

Thank the reviewer for his/her suggestion regarding Supplementary Figure 1. We have uploaded a new Supplementary Figure 1.

Round 2

Reviewer 1 Report

Comments and Suggestions for Authors

The revised manuscript is acceptable for further procedure.

Author Response

Response letter to reviewers

The authors would like to express our gratitude to the reviewers for their meticulous and thorough evaluation of our manuscript. By incorporating their valuable suggestions, we are confident that the manuscript has been improved significantly. We have made some changes, and the changes made to the original manuscript have been highlighted in the revised version of the manuscript according to the reviewer's comments.

Reviewer 2 Report

Comments and Suggestions for Authors

You have to update references according to the reviewer's suggestions and to your changes.

Comments on the Quality of English Language

Minor editing.

Author Response

Response letter to reviewers

The authors would like to express our gratitude to the reviewers for their meticulous and thorough evaluation of our manuscript. By incorporating their valuable suggestions, we are confident that the manuscript has been improved significantly. We have updated the references according to the reviewer's suggestions. All other changes made to the original manuscript have been highlighted in the revised version of the manuscript according to the reviewer's comments.